# Complete Mitogenome and Phylogenetic Analysis of the *Carthamus tinctorius* L.

**DOI:** 10.3390/genes14050979

**Published:** 2023-04-26

**Authors:** Zhihua Wu, Tiange Yang, Rui Qin, Hong Liu

**Affiliations:** 1College of Life Sciences, Zhejiang Normal University, Jinhua 321004, China; 2Hubei Provincial Key Laboratory for Protection and Application of Special Plant Germplasm in Wuling Area of China, College of Life Sciences, South-Central Minzu University, Wuhan 430074, China; yangtge@163.com (T.Y.);

**Keywords:** *Carthamus tinctorius*, mitogenome, RNA editing, horizontal transfer, phylogeny

## Abstract

*Carthamus tinctorius* L. 1753 (Asteraceae), also called safflower, is a cash crop with both edible and medical properties. We analyzed and reported the safflower mitogenome based on combined short and long reads obtained from Illumina and Pacbio platforms, respectively. This safflower mitogenome mainly contained two circular chromosomes, with a total length of 321,872 bp, and encoded 55 unique genes, including 34 protein-coding genes (PCGs), 3 rRNA genes, and 18 tRNA genes. The total length of repeat sequences greater than 30 bp was 24,953 bp, accounting for 7.75% of the whole mitogenome. Furthermore, we characterized the RNA editing sites of protein-coding genes located in the safflower mitogenome, and the total number of RNA editing sites was 504. Then, we revealed partial sequence transfer events between plastid and mitochondria, in which one plastid-derived gene (*psaB*) remained intact in the mitogenome. Despite extensive arrangement events among the three mitogenomes of *C. tinctorius*, *Arctium lappa*, and *Saussurea costus*, the constructed phylogenetic tree based on mitogenome PCGs showed that *C. tinctorius* has a closer relationship with three Cardueae species, *A. lappa*, *A. tomentosum*, and *S. costus*, which is similar to the phylogeny constructed from the PCGs of plastid genomes. This mitogenome not only enriches the genetic information of safflower but also will be useful in the phylogeny and evolution study of the Asteraceae.

## 1. Introduction

As the center of plant energy metabolism, mitochondria are involved in many aspects of plant growth and development, such as respiration, programmed cell death, and even cytoplasmic male sterility [1]. Based on the endosymbiotic theory, researchers generally believe that plant mitochondria evolved from free-living bacteria [2]. However, compared with animal mitogenomes, plant mitogenomes are more variable and show rich diversity in genome size, structure, gene content, and evolutionary rate [3]. Plant mitogenomes exist not only in ring form but also in other different forms [1]. Compared with microbial and animal mitogenomes, plant mitogenomes are larger in size and more complex in structure due to the migration of large amounts of exogenous DNA and the recombination of repetitive sequences [4,5,6]. The mitogenome has a low synonymous substitution rate compared with the plant nuclear and chloroplast genomes, and many homologous fragments in the plant mitogenomes are derived from the chloroplast and nucleus [7]. For example, in *Cucumis melo* (2.74 Mb) [8] and *Larix sibirica* (11.7 Mb) [9], which had the large plant mitogenome, many DNA homologous sequences originate from both the nuclear and chloroplast genomes.

Horizontal gene transfer (HGT) is a relatively rare phenomenon across plant evolution but has been shown to occur in some plant species [10]. Through HGT, plants can acquire new genes or functional modules that enhance their ability to adapt to their environment, help them adapt to new environments and ecological niches, accelerate their evolutionary rate and diversity, and thus improve their competitiveness and survival. Meanwhile, HGT may also bring some negative effects, such as increased gene conflicts and incompatibility, which need further research and investigation. While horizontal gene transfer from the chloroplast and nuclei to mitochondria may account for the complex structure of plant mitogenomes, repetitive sequences in plant mitogenomes may also trigger changes in the genomic size and structure [11]. Diversity in the mitogenomic size and structure of plants provides rich information on their ancestral characteristics and evolution [3].

*C. tinctorius* L. (2n = 2x = 24), also known as safflower, is an annual herb of the *Carthamus* genus in the Asteraceae family, a plant native to the Middle East and Central Asia that is now widely cultivated throughout the world. *C. tinctorius* is a cash crop mainly used for the production of red dyes and edible oils [12,13]. Its seeds contain up to 40% oil, rich in unsaturated fatty acid of linoleic acid (up to 80%), which can be used in the production of edible, medicinal, and industrial oils, among other products. Moreover, safflower is drought-tolerant in cultivation and is suitable for growing in dry conditions, but it also needs full sunlight and moderate amounts of water [14,15]. In horticulture, it can also be used as an ornamental plant, planted in flower beds, borders, or lawns to add beauty to the garden. In addition, safflower is also a traditional herbal plant rich in flavonoids, which have some antioxidant and anti-inflammatory effects and can be used to treat hypertension and coronary heart, cardiovascular, and gynecological diseases [16,17,18,19]. For example, safflower petals have uniquely flavonoid, hydroxysafflor yellow A (HSYA), which has various pharmacological functions, such as cerebral protective, myocardial, and antioxidative effects. Therefore, safflower is becoming increasingly important due to its industrial, ornamental, medicinal, and oleaginous values and is cultivated in more than 60 countries with great edaphoclimatic adaptations for the different regions.

The whole genomic sequences, including nuclear, chloroplast, and mitochondrion, would pave a solid foundation for understanding genetic features. So far, the reports on assembled and annotated plant mitogenomes in public database have been still relatively few compared with plastid genomes. Previously, our research team has de novo assembled the chromosome nuclear and chloroplast genome of safflower and revealed the potential regulatory roles of chalcone synthase (CHS) gene and fatty acid desaturase 2 (FAD2) families in the biosynthesis of HSYA and linoleic acid, respectively [20,21]. However, the safflower mitogenome is still lacking due to its complex structure. The key evolutionary features and fundamental biological mechanisms of safflower remain to be elucidated, such as the relationships between energy metabolism and drought response and the biosynthesis of flavonoids and fatty acid. Here, we report and analyze the complete safflower mitogenome, which will contribute to future molecular genetics and evolutionary studies of safflower.

## 2. Materials and Methods

### 2.1. Plant Materials, Total DNA Extraction, and Genome Sequencing

The safflower cultivar (known as Anhui-1) was cultivated in the experimental field at South-Central Minzu University, Wuhan, China, in 2017 [21]. The voucher specimens were deposited at the Herbarium of South-Central Minzu University (HSN) with the ID HSNAH001 (114°23′52.650″ E, 30°29′42.843″ N, Wuhan, https://www.scuec.edu.cn/, accessed on 20 April 2023; Liu Hong, liuhong@scuec.edu.cn). The figure of safflower used in this study can also be seen in the safflower genome database developed by our research team (https://safflower.scuec.edu.cn/, accessed on 20 April 2023).

To construct the complex and dynamic mitogenome of *C. tinctorius*, the genomic DNA was extracted using the improved CTAB method [22], and the genome was sequenced on both platforms of Illumina HiSeq 2500 and PacBio’s Single Molecule Real-Time (SMRT) in Frasergen company (Wuhan, China).

### 2.2. Assembly and Annotation of Mitogenome

We filtered the short reads using fastp [23] to obtain clean short reads. We used GetOrganelle v.1.7.5 [24] to assemble the chloroplast genome of safflower from clean short reads with the following parameters: ‘-R 15 -k 21,45,65,85,105 -F embplant_pt’. For the mitogenome, we used GetOrganelle v.1.7.5 [24] for de novo assembly and then removed chloroplast genome fragments to obtain a draft mitogenome with the specific parameters ‘-R50 -k 21,45,65,85,105,127 -F embplant_mt’. Then, minimap2 [25] was used to map the draft mitogenome to the PacBio SMRT reads (~380 GB), the long sequence fragments of mitogenome were extended using samtools v. 1.7 [26], and Canu v. 2.2 [27] was used for the correction and assembly of the long reads. Subsequently, the genomic sequence was further assembled using Flye v.2.9 [28] based on the corrected sequences from Canu v. 2.2 [27]. We evaluated and complemented the two results. Finally, two large overlapping groups were assembled into the mitogenome and formed a mitogenome with two rings.

Referring the mitogenome of *Chrysanthemum boreale* (NC_039757.1), the safflower mitogenome was annotated using the three software packages PMGA v.2022 (http://www.1kmpg.cn/mgavas/, accessed on 20 April 2023), Geneious Prime 2021.2.2 [29], and GeSeq v.2017 (https://chlorobox.mpimp-golm.mpg.de/geseq.html, accessed on 20 April 2023). Then, OGDRAW v.2019 (https://chlorobox.mpimp-golm.mpg.de/OGDraw.html, accessed on 20 April 2023) was further used to draw the safflower mitogenome map.

### 2.3. Repeats Identification and SSRs Analysis

The simple sequence repeats (SSRs), also known as microsatellites, were identified using MISA v.2.1 [30] (https://webblast.ipk-gatersleben.de/misa/, accessed on 20 April 2023). Moreover, the parameters of minimum repeat numbers were set as follows: 10, 5, 4, 3, 3, and 3 for mono-, di-, tri-, tetra-, penta-, and hexanucleotides, respectively. Additionally, REPuter v.2001 [31] (https://bibiserv.cebitec.uni-bielefeld.de/reputer/, accessed on 20 April 2023) was further used to calculate various repeats, including forward repeats, reverse repeats, palindromic repeats, and complementary repeats with the minimal repeat size of 30 bp.

### 2.4. Collinear Analysis and Rearrangement Analysis of Mitogenomes between Safflower and Two Other Asteraceae Species

To compare the similarity and collinearity between the mitogenomes of safflower and other species in Asteraceae, we performed a collinear analysis. We used the Asteraceae mitogenomes of safflower, *Arctium lappa* and *Saussurea costus*, for collinear analysis in NGenomeSyn v.1.4.1 [32] (https://github.com/hewm2008/NGenomeSyn, accessed on 20 April 2023). Then, we used mauve v.2.4.0 software [33] for genomic rearrangement analysis.

### 2.5. Analysis of Codon Usage Bias Patterns for Protein-Coding Genes (PCGs)

Relative synonymous codon usage (RSCU) and effective codon number (ENC) can be used to characterize the pattern of codon usage preference. Among them, ENC was also used to detect the effect of the base composition of genes on codon usage preference. Therefore, we used a cloud platform (http://112.86.217.82:9919/#/tool/alltool/detail/214, accessed on 20 April 2023) to analyze the RSCU and ENC for mitogenomic PCGs [34].

### 2.6. Identification of RNA Editing Sites 

To further count the RNA editing sites of PCGs in the safflower mitogenome, we used an online cloud platform (http://112.86.217.82:9919/#/tool/alltool/detail/336, accessed on 20 April 2023) to predict the possible RNA editing sites in the safflower mitochondrial genome, setting a threshold value of C = 0.2. A lower threshold value will predict more complete RNA editing sites but may also increase the probability of misidentification.

### 2.7. Intracellular Gene Transfer Analysis

To identify potential horizontal transfer events that occurred between the chloroplast and mitogenome of safflower, we compared the safflower chloroplast genome (MK983238.1) and mitogenome (OQ621745 and OQ621746) using BLASTn program. The potential transferred DNA sequences were extracted by their genomic position and analyzed and further visualized using Circoletto v.07.09.16 (http://tools.bat.infspire.org/tools/circoletto/, accessed on 20 April 2023). 

### 2.8. Phylogenetic Analysis

To understand the phylogeny of safflower in the plant kingdom, a total of 26 complete mitochondrial genomes from different plants were obtained from the GenBank database with *Arabidopsis thaliana* as the outgroup. We used PhyloSuite v. 1.2.2 [35] to extract common PCGs from both the mitogenomes and chloroplast genomes of these species. Each PCG sequence was then aligned using MAFFT v. 7.4 with default parameters [36]. After the manual corrections, the aligned sequences for each species were further concatenated using PhyloSuite v. 1.2.2 [35]. Based on the matrix of concatenated sequences, the maximum-likelihood (ML) tree was constructed using IQ-TREE v. 2.1.2 [37], and modelfinder was used to find the best-fit model as TVM+F+R2. For the Bayesian inferred (BI) tree, we used PhyloSuite v. 1.2.2 [35] to find the best-fit model as GTR and then used MrBayes v. 3.2.6 software to perform the analysis with the following parameters: ngen = 1,000,000, nruns = 2, nchains = 4, samplefreq = 1000, nst = 6, and rates = invgamma. Then, the first 25% of the trees were discarded as burn-in. Tree visualization was achieved in Figtree v. 1.4.3 (https://github.com/rambaut/figtree/releases, accessed on 20 April 2023). Finally, the support and posterior probabilities on the species branch nodes are manually integrated according to the topology of the ML and BI trees.

## 3. Results and Discussion

### 3.1. Mitogenome Characterization of C. tinctorius

Most plant mitogenomes mainly exist as linear and partially circular DNA [1]. The safflower mitogenome was assembled as two circular molecules (chromosome 1: 258,555 bp, chromosome 2: 63,317 bp) with a total length of 321,872 bp (https://note.youdao.com/s/7DCcbNy8, accessed on 20 April 2023; https://safflower.scuec.edu.cn/, accessed on 20 April 2023; GenBank accession number: OQ621745 of chromosome 1, OQ621746 of chromosome 2) (Figure 1; Table 1). There were 55 unique genes predicted and annotated in the mitogenome, including 34 PCGs, 18 tRNA genes, and 3 rRNA genes (Table 2). Among the 34 conserved PCGs in the mitogenomes, 18 genes code for electron transport proteins and ATP synthase, including 8 subunits of complex I (*nad1*, *nad2*, *nad3*, *nad4*, *nad4L*, *nad5*, *nad6*, *nad7*, and *nad9*), 1 subunit of complex III (*cob*), 3 subunits of complex IV (*cox1*, *cox2*, and *cox3*), and 5 subunits of complex V (*atp1*, *atp4*, *atp6*, *atp8*, and *atp9*). Furthermore, there were three genes for large ribosomal proteins (*rpl5*, *rpl10*, and *rpl16*), five genes for small ribosomal proteins (*rps1*, *rps3*, *rps4*, *rps12*, and *rps13*), four genes for cytochrome c biogenesis (*ccmB*, *ccmC*, *ccmFC*, and *ccmFN*), one gene for succinate dehydrogenase subunit (*sdh4*), one gene for photosystem I apoprotein (*psaB*), and two genes for the maturase and transport membrance protein (*matR* and *matB*). 

The total size of the 34 PCGs was 32,352 bp, accounting for 10.05% of the safflower mitogenome. Each PCG varied from 261 bp (*atp9*) to 2253 bp (*psaB*) in length. Except for the *mttB* and *nad3* genes using TTG as the starting codon, other PCGs used ATG as the starting codon. Moreover, eight genes having an intron structure were discovered, and they were *nad1*, *nad2*, *nad4*, *nad5*, *nad7*, *ccmFC*, *cox2*, and *rps3*. Furthermore, three trans-spliced introns were observed in *nad1*, *nad2*, and *nad5* in different chromosomal coding regions. In addition, most of the total identified 18 tRNA genes had only one copy, except for *trnQ-UUG* with two copies, as well as both *trnM-CAU* and *trnS-GCU* with three copies. Furthermore, a total of 241 open reading frames (ORFs), accounting for 35.29% (113,592 bp) of the safflower mitogenome, were identified with the sequence length greater than 100 using Geneious Prime. Most of these function-unknown ORFs had one copy with a variety of length from 300 to 2757 bp, among which 14 ORFs were greater than 1000 bp. 

In this study, our reported safflower mitogenome was similar to most Asteraceae plants in terms of the sequences of coding and noncoding regions. We further comparatively analyzed the basic features of the mitogenomes of safflower and other Asteraceae species and found that safflower differed from other closely related species in the presence of a nearly complete *psaB* gene in the safflower mitogenome. The *psaB* gene is mainly present in the chloroplast genome, and the existence in the safflower mitogenome may be related to the adaptive evolution of safflower. In addition, eight intron-containing genes were also identified in the safflower mitogenome, including *ccmFC*, *cox2*, *nad1*, *nad2*, *nad5*, *nad7*, and *rps3*. Meanwhile, the positions and splicing styles of introns in the eight genes showed high similarity to those in the other Asteraceae mitogenomes [38,39,40].

### 3.2. Repeats, SSRs, and Collinear Analysis in the C. tinctorius Mitogenome

Repeats are an important data source for developing molecular markers for population analysis. They include tandem, short, and large repeats and are extensively present in the mitochondrial genomes [41,42,43]. Mitogenome repeats are often critical for intermolecular recombination, which can alter the genome size and generate genomic structural variation [3,44]. A total of 47 repeats were identified in the safflower mitogenome, ranging from 31 to 15,273 bp in length (Figure 2a; Appendix A). Two types of repeats, including palindromic and forward repeats, were mainly detected in the safflower mitogenome. Of them, three large repeats (R1–R3) were greater than 1000 bp, while three relatively small repeats (R4–R6) varied from 104 to 734 bp. Most large repeats contained two copies of each repeat. In addition, we found a total of 137 SSRs presented in the safflower mitogenome, with 22 mononucleotides, 36 dinucleotides, 13 trinucleotides, 53 tetranucleotides, 9 pentanucleotides, and 4 hexanucleotides, with the largest proportion of tetranucleotides up to 38.69% (Figure 2b). In addition, according to the collinearity analysis result (Figure 3 and Appendix A), there were obvious intragenomic rearrangement events among the different mitogenomes. There was no obvious synteny observed between the mitogenomes of *C. tinctorius*, *A. lappa*, and *S. costus*.

In total, these repeats accounted for 7.75% of the safflower mitogenome. In genomes, repetitive sequences are common, and duplications can often lead to changes in the size of the genome. Especially in angiosperms, it is commonly used to explain the diversity of the mitochondrial genome size and structure [5]. However, these large repeats (>1000 bp) account for only 6.70% (21,579 bp) of the safflower mitogenome, which may not be the major cause of the expanded mitogenome size in safflower. Rearrangements of plant mitogenomes were relatively common, and the mitogenome structure was not as stable as that of the chloroplast genome probably due to the abundance of repetitive sequences and the complex genome structure in mitogenomes.

### 3.3. Codon Usage Bias Patterns and RNA Editing Sites Analysis for Protein-Coding Genes (PCGs)

To investigate the codon preferences of protein-coding genes in the safflower mitogenome, we performed ENC-plot, PR2-plot, and RSCU analyses. ENC-plot analysis provides a visual indication of the extent to which codon usage patterns and preference formation are affected by natural mutation and selection. Theoretical ENC values vary from 20 to 61, with values closer to 20 indicating that the gene is less influenced by natural selection and vice versa [34]. The results of the ENC-plot analysis of the safflower mitogenome showed that most of the genes were some distance away from the standard curve, and the actual ENC values were somewhat different from the expected values, indicating that the effect of natural selection on the codons was greater. However, some genes were located around the standard curve, and the actual ENC values were basically similar to the expected values, indicating that mutations had a greater effect on these codons (Appendix A, Appendix A). If the genes within the PR2-plot are evenly distributed, it indicates that the A, T, C, and G of each base in the codon are used with the same frequency. The results of the PR2-plot analysis of the safflower mitogenome showed an irregular distribution of mitochondrial genes, most of which were distributed in the lower part of the plot (Appendix A, Appendix A). The different frequencies of G and C, A, and T suggested that the bias in the use of codons in the safflower mitogenome may be influenced by several factors.

After analyzing 10,784 codons (including terminated codons) encoding 34 safflower mitochondrial PCGs, we found that the RSCU values of all codons except 34 codons (Ala (GCG, GCC, GCA), Cys (UGC), Glu (GAG), Phe (UUC), Asp (GAC), Gly (GGC, GGG), Lys (AAG), His (CAC), Val (GUC, GUG), Trp (UGG), Ile (AUC, AUA), Leu (CUG, CUC, CUA), Met (CUG, UUG), Asn (AAC), Arg (CGC, CGG, AGG), Pro (CCG, CCC), Gln (CAG), Ser (AGC, UCG, UCC), Thr (ACG, ACA), and Tyr (UAC))) were higher than 1.0. This phenomenon was also commonly found in the mitochondria of most Asteraceae species. Of the 10,784 codons, 6660 codons (61.76%) had RSCU values above 1.0, indicating that they were used more frequently than other codons (Figure 4a; Appendix A). The codon usage frequencies of mitogenomes of four Cardueae species (*C. tinctorius*, *A. lappa*, *A. tomentosum*, and *S. costus*) suggested that the number of codons with RSCU over 1 in these four species ranged from 5960 to 5970 (Appendix A). The most commonly used codons in these four Cardueae plants were GCU (Ala), UAU (Tyr), and CAU (His), all of which had RSCU values over 1.5. RNA editing is a widespread phenomenon in mitochondria, chloroplast, and three RNA-coding genes (mRNA, rRNA, and tRNA) of the nucleus. RNA editing events involve in the additional insertion of bases or substitutions of a single base, resulting in the occurrence of the start or stop codons [45]. Here, we performed further detection of RNA editing sites in 34 mitochondrial PCGs, of which 31 mitochondrial PCGs had RNA editing sites (Figure 4b; Appendix A). In the safflower mitochondrial PCG, we identified a total of 504 RNA editing sites, of which the genes with more than 30 RNA editing sites were *ccmB*, *ccmC*, *ccmFN*, *mttB*, and *nad7*. In addition, 14 kinds of RNA editing sites were also detected from PCGs in the safflower mitochondrial genome. As for editing efficiency, we found that these mitochondrial PCGs with editing efficiencies higher than 80% were predominated (Appendix A).

### 3.4. Intracellular Gene Transfer of C. tinctorius Organelle Genomes

The foreign DNA insertion into the mitogenome, including intracellular gene transfer (IGT) of nuclear and chloroplast DNA or the horizontal gene transfer (HGT) of other mitochondrial DNA, is a frequent phenomenon in the plant kingdom. Both IGT and HGT events contribute to the genomic expansion of mitochondria, eventually resulting in the adaptive evolution of mitochondria. Here, to explore the IGT occurrence between the safflower chloroplast and mitochondria, we identified chloroplast-derived sequences in the safflower mitogenome using BLASTn. In total, 26 similar sequence fragments were identified, all of which are located on chromosome 1, which had high nucleotide sequence identity with the corresponding chloroplast sequence of safflower (Figure 5a). The total length of these chloroplast-derived fragments was 11,156 bp, occupying a proportion of 3.47% of the safflower mitogenome. The length of chloroplast-derived fragments varied from 28 to 2556 bp (Appendix A). Meanwhile, some chloroplast-derived genes, such as the *psaB* and tRNA-related genes, mainly functioned in the processes of photosynthesis, transcription, and translation. These results suggested that extensive IGT events occurred between the safflower mitogenome and chloroplast genome.

The IGT events between organelle genomes were relatively common in most terrestrial plants. However, the number of transferred sequences and sequence lengths extremely varied among the diverse plant species [46,47]. For example, about 0.35% of chloroplast sequences were transferred to the mitogenome in safflower, which was much less than the numbers of chloroplast DNA sequences in the mitogenomes of some plants (3–6%) [48]. In addition, most of the chloroplast-derived sequences were partially non-functional gene fragments with the exceptions for some tRNA genes, which was in agreement with most other terrestrial plants [49]. The almost complete chloroplast-derived *psaB* gene was identified in the safflower mitogenome (Figure 5b), almost identical to the *psaB* gene in the chloroplast, which was also frequently found in the mitogenomes of most angiosperms. Therefore, the identification of mutually transferred DNA fragments between safflower organelle genomes can help us further understand the evolution of plants.

### 3.5. Phylogenetic Analysis

To analyze the phylogeny of safflower in Asteraceae species, the PCGs of the chloroplast and mitochondrial genome were used for phylogenetic analysis. In total, we selected 27 angiosperm species, including 26 Asteraceae species, and *A. thaliana* as an outgroup. Phylogenetic trees were built based on Bayesian inference (BI) and maximum-likelihood (ML) methods using common PCGs in the chloroplast and mitochondrial genomes, respectively. We found that our phylogenetic results were in general agreement with the previous phylogenetic position of safflower. Our phylogenetic results showed that safflower was more closely related to the three Cardueae species, *A. lappa*, *A. tomentosum*, and *S. costus* (Figure 6). Our phylogenetic results were almost consistent with those of previous studies, indicating that the phylogenetic trees based on the mitochondrial PCGs were reliable like chloroplast PCGs. Hence, the safflower mitogenome will provide a reference for future phylogenetic studies of other Asteraceae [20,40].

## 4. Conclusions

The topological structure of the plant mitogenome is highly dynamic. It has a diversity of conformations, including linear, circular, and branched structures. Usually, a main circle containing all mitochondrial genes represents the main plant mitogenome. In this study, we analyzed and reported the safflower mitogenome, which mainly contained two circular molecular sequences with a total length of 321,872 bp, encoding 55 different genes. In the safflower mitogenome, there were 24,953 bp of repeats over 30 bp in length, accounting for 7.75% of the whole genome. We compared the chloroplast genome and mitogenome of safflower and found partial sequence transfer events, with one gene from the chloroplast (*psaB*) remaining intact in the mitogenome. Although there were extensive genome rearrangement events between the mitogenomes of *C. tinctorius*, *A. lappa*, and *S. costus*, phylogenetic trees based on the organelle genomes support that safflower was more closely related to *A. lappa*, *A. tomentosum*, and *S. costus*. This safflower mitogenome not only enriched the genetic information but also contributed to the phylogenetic and evolutionary studies of Asteraceae.

## Figures and Tables

**Figure 1 genes-14-00979-f001:**
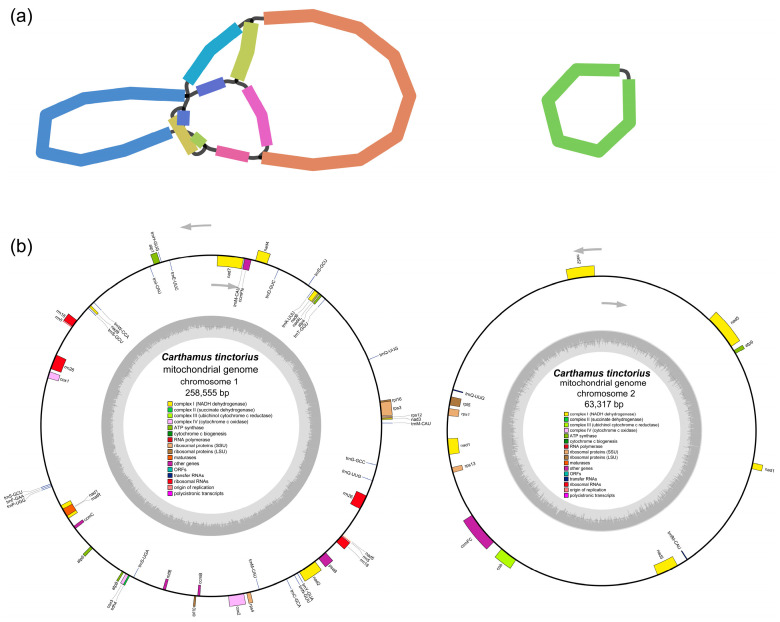
Mitogenome of *C. tinctorius*. (**a**) Topological structures of the mitochondrial contigs of *C. tinctorius* displayed in Bandage. Different line segments represent different contigs. (**b**) Two circular mitochondrial DNA maps. Genes outside the circles were transcribed counterclockwise, while those inside the circles were transcribed clockwise. The light and dark gray shadows inside the inner circle indicated AT and GC content, respectively.

**Figure 2 genes-14-00979-f002:**
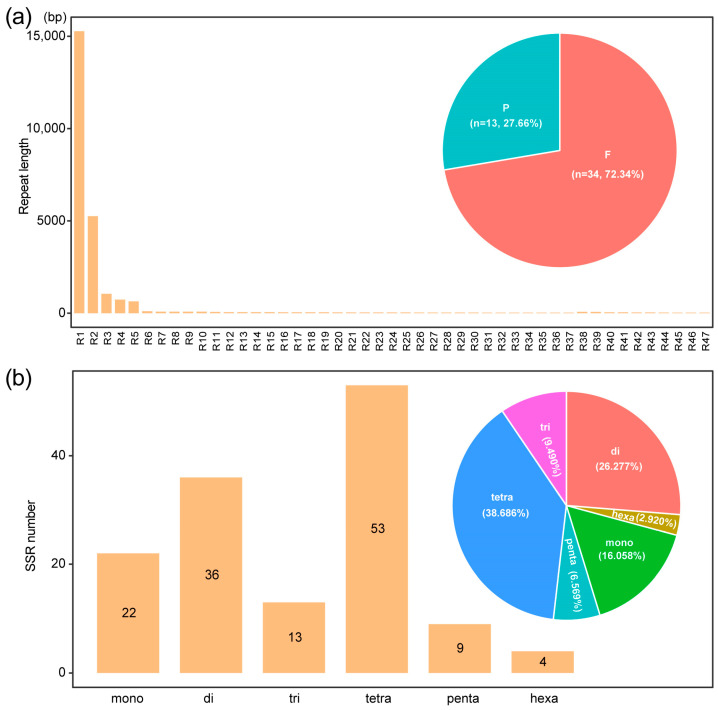
Repeat sequence analysis of safflower mitogenome. (**a**) Length and percentage of repeats. P indicated palindromic repeats; F indicates forward repeats. (**b**) Number and percentage of SSRs in safflower mitogenome. Mono, di, tri, tetra, penta, and hexa represented mononucleotide, dinucleotide, trinucleotide, tetranucleotide, pentanucleotide, and hexanucleotide, respectively.

**Figure 3 genes-14-00979-f003:**
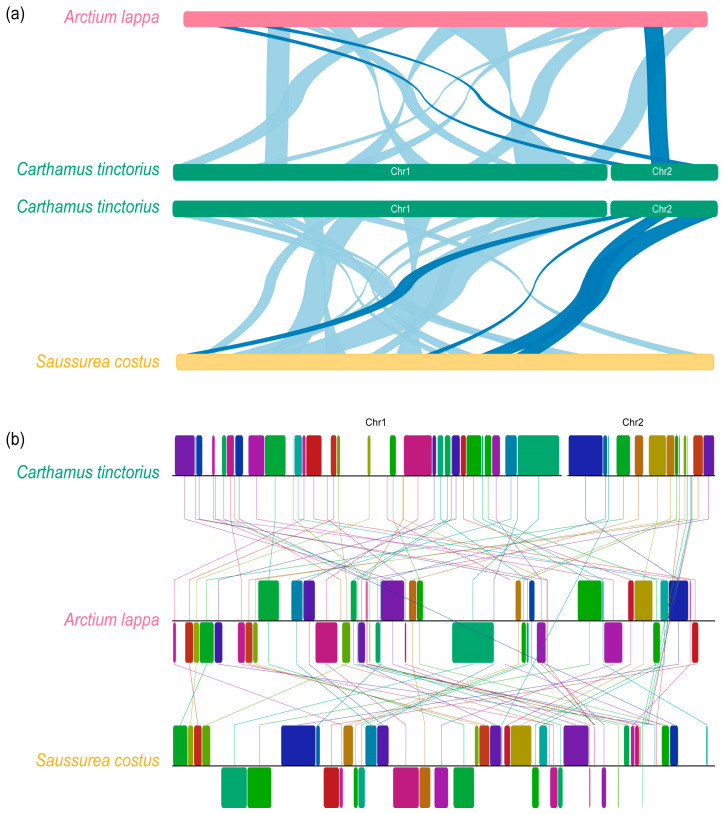
Collinearity analysis and genomic rearrangement analysis between *C. tinctorius* mitogenome and *Arctium lappa* and *Saussurea costus* mitogenomes. (**a**) Collinearity analysis of large segments of mitogenome using NGenomeSyn. (**b**) Mitogenomic rearrangement analysis using mauve.

**Figure 4 genes-14-00979-f004:**
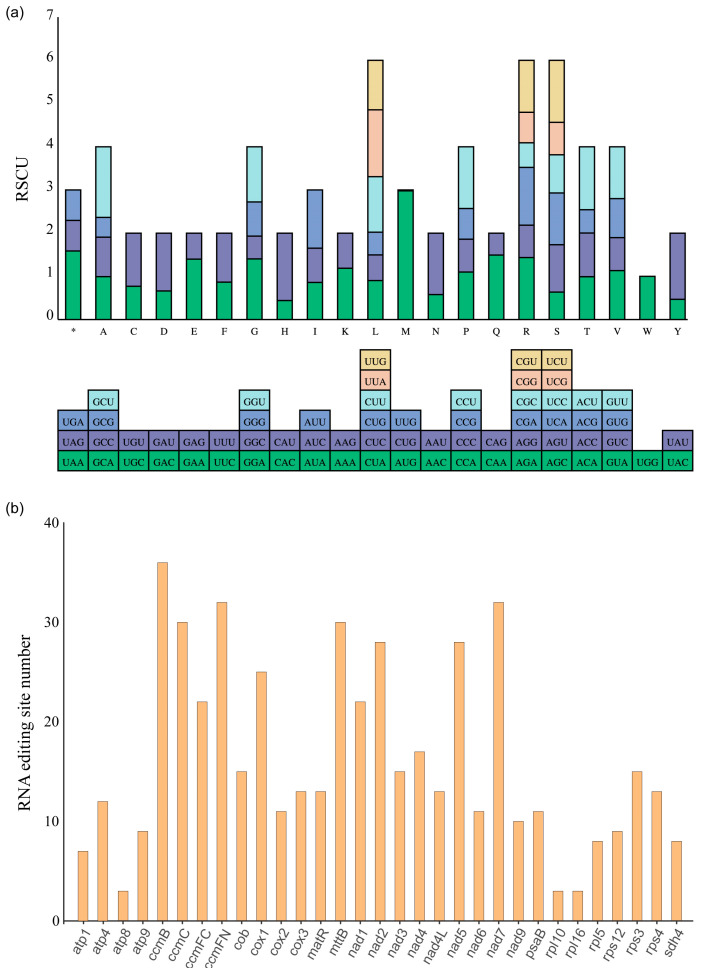
Relative synonymous codon usage (RSCU) and RNA editing analysis. (**a**) RSCU values of PCGs in *C. tinctorius* mitogenome. X-axis showed different kinds of codon families for each amino acid. RSCU values represented frequency for specific codon in comparison with uniform synonymous codon usage. Asterisk indicated termination codon. (**b**) Statistics of RNA editing sites in *C. tinctorius* mitogenome.

**Figure 5 genes-14-00979-f005:**
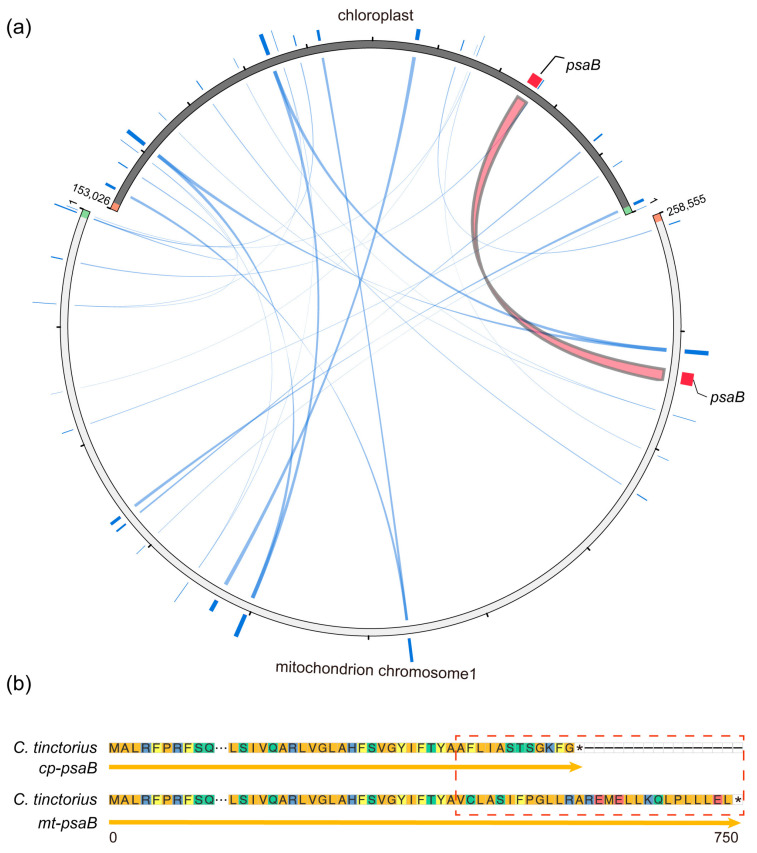
Analysis of gene transfer between mitogenome and chloroplast genome of safflower. (**a**) Schematic diagram of chloroplast DNA fragments transferred into safflower mitogenome. Red line represented transferred DNA fragment having 100% similarity, orange line represented DNA fragment with 75–99% similarity, green line represented DNA fragment with 50–75% similarity, and blue line represented DNA fragment similarity less than 50%. (**b**) Comparison of *psaB* gene between safflower chloroplast and mitochondrial genomes. Multiple sequence alignment was performed using MEGA11, followed by multiple sequence alignment and visualization using ggmsa in R environment. Ellipses in middle represented same part. Red dotted box represented differential region.

**Figure 6 genes-14-00979-f006:**
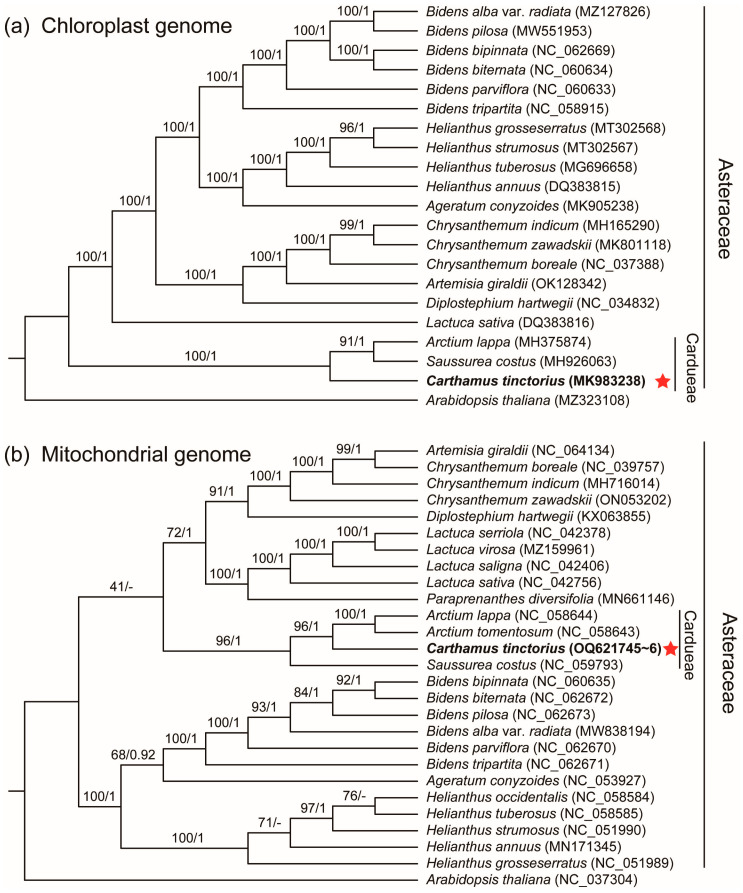
Phylogenetic relationships among *C. tinctorius* using Bayesian inference (BI) and maximum-likelihood (ML) methods. (**a**) Phylogenetic tree based on chloroplast PCGs. (**b**) Phylogenetic tree based on mitochondrial PCGs. Numbers on branches indicated bootstrap values of ML tree (left) and posterior probabilities of BI tree (right), respectively. Red stars and bold denote the safflower in this study.

**Table 1 genes-14-00979-t001:** Statistics of length, GC content, and predicted genes of two chromosomes in *C. tinctorius* mitogenome.

Sequences	Chromosome 1	Chromosome 2
Length (bp)	258,555	63,317
Gene name	*atp1*, *atp4*, *atp6*, *atp8*, *ccmB*, *ccmC*, *ccmFN*, *cox1*, *cox2*, *cox3*, *matR*, *mttB*, *nad1*, *nad2*, *nad3*, *nad4*, *nad4L*, *nad4*, *nad5*, *nad6*, *nad7*, *nad9*, *psaB*, *rpl10*, *rpl16*, *rps12*, *rps3*, *rps4*, *rrn18*, *rrn26*, *rrn5*, *sdh4*, *trnC-GCA*, *trnD-GUC*, *trnE-UUC*, *trnF-GAA*, *trnG-GCC*, *trnH-GUG*, *trnI-CAU*, *trnK-UUU*, *trnM-CAU*, *trnN-GUU*, *trnP-UGG*, *trnQ-UUG*, *trnS-GCU*, *trnS-UGA*, *trnT-GGU*, *trnW-CCA*, *trnY-GUA*	*atp9*, *ccmFC*, *cob*, *nad1*, *nad2*, *nad5*, *rpl5*, *rps1*, *rps13*, *trnQ-UUG*, *trnfM-CAU*
Gene number	48	11
GC content	45.24%	45.43%

**Table 2 genes-14-00979-t002:** Functional annotation of genes in *C. tinctorius* mitogenome.

Functional Classification of Genes	Gene Name
ATP synthase	*atp1*, *atp4*, *atp6*, *atp8*, *atp9*
Cytochrome c biogenesis	*ccmB*, *ccmC*, *ccmFc* *, *ccmFn*
Ubichinol cytochrome c reductase	*cob*
Cytochrome c oxidase	*cox1*, *cox2* **, *cox3*
Maturases	*matR*
Transport membrance protein	*mttB*
NADH dehydrogenase	*nad1* **, *nad2* ***, *nad3*, *nad4* *, *nad4L*, *nad5* **, *nad6*, *nad7* ****, *nad9*
Ribosomal proteins (LSU)	*rpl10*, *rpl16*, *rpl5*
Ribosomal proteins (SSU)	*rps12*, *rps3* *, *rps4*
Succinate dehydrogenase	*sdh4*
Ribosomal RNAs	*rrn18* (×2), *rrn26* (×2), *rrn5* (×2)
Transfer RNAs	*trnC-GCA*, *trnD-GUC*, *trnE-UUC*, *trnF-GAA*, *trnG-GCC*, *trnH-GUG*, *trnK-UUU*, *trnM-CAT*, *trnM-CAU* (×3), *trnN-GUU*, *trnP-UGG*, *trnQ-UUG* (×2), *trnS-GCU* (×3), *trnS-UGA*, *trnT-GGU*, *trnT-UGU*, *trnW-CCA*, *trnY-GUA*
Photosystem I	*psaB*

Note: Gene *: One-intronic gene; Gene **: Two-intronic gene; Gene ***: Three-intronic gene; Gene ****: Four-intronic gene; Gene (×2, ×3): Copy number for multi-copy genes.

## Data Availability

The genomic data that support the findings of this study were available in GenBank of NCBI (http://www.ncbi.nlm.nih.gov/, accessed on 20 April 2023) under the accession no. OQ621745-OQ621746. The associated BioProject, BioSample, and SRA numbers are PRJNA642978, SAMN15402361, and SRR12189953-SRR12189989 of Pacbio SMRT, SRR12142474, and SRR12160977 of Illumina HiSeq, respectively.

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
