# Peer review of "Complete Mitogenome and Phylogenetic Analysis of the Carthamus tinctorius L."

_genes, 2023, doi:10.3390/genes14050979_

Round 1

Reviewer 1 Report

In the paper Wu et al., "Complete mitochondrial genome sequence and comparative 2 analysis of the Carthamus tinctorius L.", authors have analyzed the mitogenome of safflower.

Comments-

1. The methods have not been adequately explained.

2. Important citations has been missed. For eg https://www.tandfonline.com/doi/abs/10.3109/19401736.2015.1018217?journalCode=imdn21

3. Figure legends needs to be explained more.

Author Response

Dear Reviewers:

Thank you for reviewing our paper and providing valuable comments and suggestions. We have carefully read your review comments and are responding to them here. Questions are indicated by Q and responses are indicated by A. Modified sections in the article are shown in yellow.

Reviewer 1

Q1. The methods have not been adequately explained.

Q2. Important citations has been missed. For eg https://www.tandfonline.com/doi/abs/10.3109/19401736.2015.1018217?journalCode=imdn21

Q3. Figure legends needs to be explained more.

A: Thank you very much for your comments. We have revised the article in its entirety, including materials and methods, references, and figure legends.

Reviewer 2 Report

Overall, the article on the complete mitochondrial genome sequence and comparative analysis of the Carthamus tinctorius L. is well-written and presented. 

The simple summary is missing.

The ‘extensive arrangement events among mitogenomes’- is that the gene rearrangement? If so, these should be well described inside the text (results and discussion).

In Materials and methods, please provide relevant citations in the 2.4 & 2.5 sections.

What is the Asterix indicated in Figure 4 (a).

The authors used two different methods (ML and BA) for phylogenetic inference. Both bootstrap values and posterior probabilities were mentioned with each node, however, I was surprised about the similar cladograms in both the ML and BA tests. Please incorporate in Supplementary materials. Further how the authors test the suitable models (any partitions of genes) for phylogenetic analyses please mentioned.

Have any patents been filed from this research (5. Patents)? Please mentioned.

Thank you.

Author Response

Dear Reviewers:

Thank you for reviewing our paper and providing valuable comments and suggestions. We have carefully read your review comments and are responding to them here. Questions are indicated by Q and responses are indicated by A. Modified sections in the article are shown in yellow.

Reviewer 2

Q1: The simple summary is missing.

A1: We have made changes.

Q2: The ‘extensive arrangement events among mitogenomes’- is that the gene rearrangement? If so, these should be well described inside the text (results and discussion).

A2: Thanks for your question. The rearrangements in this study were intra-genomic rearrangements. Rearrangements of plant mitogenomes were relatively common, and the mitogenome structure was not as stable as that of chloroplast genome, probably due to the abundance of repetitive sequences and the complex genome structure in mitogenomes.

Q3: In Materials and methods, please provide relevant citations in the 2.4 & 2.5 sections.

A3: Thanks for your suggestion. We have added the corresponding literature.

Q4: What is the Asterix indicated in Figure 4 (a).

A4: OK. We have added to the article figure legend that the asterisk indicates the termination codon.

Q5: The authors used two different methods (ML and BA) for phylogenetic inference. Both bootstrap values and posterior probabilities were mentioned with each node, however, I was surprised about the similar cladograms in both the ML and BA tests. Please incorporate in Supplementary materials. Further how the authors test the suitable models (any partitions of genes) for phylogenetic analyses please mentioned.

A5: Thanks for your question. For ML tree construction we used IQTree v2.1.2 software and modelfinder for best-fit model finding. for BI tree construction we used MrBayes v3.2.6 and Phylosuite for best-fit model finding. For chloroplast PCGs trees, the topologies obtained using ML and BI methods are consistent. However, for mitochondrial PCGs trees, there are some branching inconsistencies between the two methods. In this study, we integrate the branching structure based on the ML tree. In this study, we integrate the support and posterior probabilities of branch nodes based on the topology of ML and BI trees. The details have been added in the section of materials and methods.

Q6: Have any patents been filed from this research (5. Patents)? Please mentioned.

A6: We did not apply for a patent.

Reviewer 3 Report

The study was focused on complete mitochondrial genome sequence and comparative analysis of the Carthamus tinctorius L. The Authors revealed that the safflower mitogenome mainly contained two circular chromosomes, with a total length of 321,872 bp, and encoded 55 unique genes, including 34 protein-coding genes, 3 rRNA genes, and 18 tRNA genes. It has been characterized the RNA editing sites of protein-coding genes located in the safflower mitogenome, the total number of RNA editing sites was 504. It was also found extensive sequence transfer events between the two organelles of plastid and mitochondria, where one plastid-derived gene (psaB) remained intact in the mitogenome.

In my opinion, the manuscript is well-organized, and the results represent a high scientific value. However, I recommend the followings improvements:

-        Discussion part od the manuscript should be extended, and more in-depth described,

-        The list of references (especially the newer ones) should be added,

-        Moderate changes of the English style and grammar are recommended.

Author Response

Dear Reviewers:

Thank you for reviewing our paper and providing valuable comments and suggestions. We have carefully read your review comments and are responding to them here. Questions are indicated by Q and responses are indicated by A. Modified sections in the article are shown in yellow.

Reviewer 3

Q1: Discussion part of the manuscript should be extended, and more in-depth described,

A1: Thank you very much for your comments. We have described the results and discussion section in more detail.

Q2: The list of references (especially the newer ones) should be added,

A2: We have added to the relatively new literature.

Q3: Moderate changes of the English style and grammar are recommended.

A3: We have optimized the article statements.

Thank you again for your review comments and suggestions, which we will take seriously and further improve our paper. If you have other comments and suggestions, please feel free to contact us. Thank you!

Sincerely yours

Zhihua Wu

Zhejiang normal university